# PARAMETRIC MANIFOLD LEARNING VIA SPARSE MULTIDIMENSIONAL SCALING

## ABSTRACT

We propose a metric-learning framework for computing distance-preserving maps that generate low-dimensional embeddings for a certain class of manifolds. We employ Siamese networks to solve the problem of least squares multidimensional scaling for generating mappings that preserve geodesic distances on the manifold. In contrast to previous parametric manifold learning methods we show a substantial reduction in training effort enabled by the computation of geodesic distances in a farthest point sampling strategy. Additionally, the use of a network to model the distance-preserving map reduces the complexity of the multidimensional scaling problem and leads to an improved non-local generalization of the manifold compared to analogous non-parametric counterparts. We demonstrate our claims on point-cloud data and on image manifolds and show a numerical analysis of our technique to facilitate a greater understanding of the representational power of neural networks in modeling manifold data.

## 1 INTRODUCTION

The characterization of distance preserving maps is of fundamental interest to the problem of non-linear dimensionality reduction and manifold learning. For the purpose of achieving a coherent global representation, it is often desirable to embed the high-dimensional data into a space of low dimensionality while preserving the metric structure of the data manifold. The intrinsic nature of the geodesic distance makes such a representation depend only on the geometry of the manifold and not on how it is embedded in ambient space. In the context of dimensionality reduction this property makes the resultant embedding meaningful.

The success of deep learning has shown that neural networks can be trained as powerful function approximators of complex attributes governing various visual and auditory phenomena. The availability of large amounts of data and computational power, coupled with parallel streaming architectures and improved optimization techniques, have all led to computational frameworks that efficiently exploit their representational power. However, a study of their behavior under geometric constraints is an interesting question which has been relatively unexplored.

In this paper, we use the computational infrastructure of neural networks to model maps that preserve geodesic distances on data manifolds. We revisit the classical geometric framework of multidimensional scaling to find a configuration of points that satisfy pairwise distance constraints. We show that instead of optimizing over the individual coordinates of the points, we can optimize over the *function that generates these points* by modeling this map as a neural network. This makes the complexity of the problem depend on the number of parameters of the network rather than the number of data points, and thus significantly reduces the memory and computational complexities, a property that comes into practical play when the number of data points is large. Additionally, the choice of modeling the isometric map with a parametric model provides a straightforward out-of-sample extension, which is a simple forward pass of the network.

We exploit efficient sampling techniques that progressively select *landmark points* on the manifold by maximizing the spread of their pairwise geodesic distances. We demonstrate that a small amount of these *landmark points* are sufficient to train a network to generate faithful low-dimensional embeddings of manifolds. Figure 1 provides a visualization of the proposed approach. In the interest of gauging their effectiveness in representing manifolds, we perform a numerical analysis to measure the quality of embedding generated by neural networks and associate an order of accuracy to a given

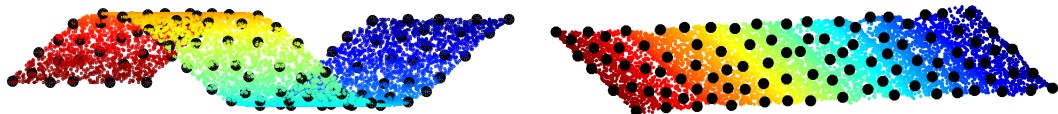

Figure 1: ***Learning to unfurl a ribbon:*** A three dimensional Helical Ribbon and its two dimensional embedding learned using a two-layer MLP. The network was trained using estimated pairwise geodesic distances between **only 100** points (marked in black) out of the total 8192 samples.

architecture. Finally, we demonstrate that parametric models provide better non-local generalization as compared to extrapolation formulas of their non-parametric counter parts.

We advocate strengthening the link between axiomatic computation and parametric learning methodologies. Existing MDS frameworks use a geometrically meaningful objective in a cumbersome non-parametric framework. At the other end, learning based methods such as DrLim Hadsell et al. (2006) use a computationally desirable infrastructure yet a geometrically suboptimal objective requiring too many examples for satisfactory manifold learning. The proposed approach can be interpreted as taking the middle path by using a computationally desirable method of a parametric neural network optimized by the geometrically meaningful cost of multidimensional scaling.

## 2 BACKGROUND AND PRIOR WORK

### 2.1 MANIFOLD LEARNING

The literature on manifold learning is dominated by spectral methods that have a characteristic computational template. The first step involves the computation of the k-nearest neighbors of all $N$ data points. Then, an $N \times N$ square matrix is populated using some geometric principle which characterizes the nature of the desired low dimensional embedding. The eigenvalue decomposition of this matrix is used to obtain the low-dimensional representation of the manifold. Laplacian Eigenmaps Belkin & Niyogi (2003), LLE Roweis & Saul (2000), HLLE Donoho & Grimes (2003), Diffusion Maps Coifman et al. (2005) etc. are considered to be local methods, since they are designed to minimize some form of local distortion and hence result in embeddings which preserve locality. Methods like Schwartz et al. (1989); Wolfson & Schwartz (1989) and Isomap Tenenbaum et al. (2000) are considered global because they enforce preserving all geodesic distances in the low dimensional embedding. Local methods lead to sparse matrix eigenvalue problems and hence are computationally advantageous. However, global methods are more robust to noise and achieve globally coherent embeddings, in contrast to the local methods which can sometimes lead to excessively clustered results. All spectral techniques are non-parametric in nature and hence do not characterize the map that generates them. Therefore, the computational burden of large spectral decompositions becomes a major drawback when the number of data-points is large. Bengio et al. (2004) and De Silva & Tenenbaum (2004) address this issue by providing formulas for out-of-sample extensions to the spectral algorithms. However, these interpolating formulations are computationally inefficient and exhibit poor non-local generalization of the manifold Bengio et al. (2013).

### 2.2 MULTIDIMENSIONAL SCALING

Multidimensional scaling (henceforth MDS) is a classical problem in geometry processing and data science and a powerful tool for obtaining a global picture of data when only pairwise distances or dissimilarity information is available. The core idea of MDS is to find an embedding $\mathbf{X} = \begin{bmatrix} \mathbf{x_1}, \mathbf{x_2}, \mathbf{x_3}...\mathbf{x_N} \end{bmatrix}$ such that the pairwise distances measured in the embedding space are faithful to the desired distances $\mathbf{D_s} = [\mathbf{d_{ij}^2}]_{i,j=1}^{i,j=N}$ as much as possible. There are two prominent versions of MDS: Classical Scaling and Least Squares Scaling. Classical Scaling is based on the observation that the double centering of a pairwise squared distance matrix gives an inner-product matrix which can be factored to obtain the desired embedding. Therefore, if $\mathbf{H} = \mathbf{I} - \frac{1}{n}\mathbb{1}\mathbb{1}^T$ is the centering

matrix, classical scaling minimizes the Strain of the embedding configuration $\mathbf{X}$ given by

$$Strain(\mathbf{X}) = ||\mathbf{X}\mathbf{X}^T + \frac{1}{2}\mathbf{H}\mathbf{D_s}\mathbf{H}||_F^2, \qquad (1)$$

and is computed conveniently using the eigen-decomposition of the $N \times N$ matrix $-\frac{1}{2}\mathbf{H}\mathbf{D_s}\mathbf{H}$. At the other end, least squares scaling is based on minimizing the misfits between the pairwise distances of $\mathbf{X} = \begin{bmatrix} \mathbf{x_1}, \mathbf{x_2}, \mathbf{x_3}...\mathbf{x_N} \end{bmatrix}$ and desired distances $[\mathbf{d_{ij}}]_{i,j=1}^{i,j=N}$ measured by the Stress function

$$Stress(\mathbf{X}) = \sum_i \sum_j (||\mathbf{x_i} - \mathbf{x_j}|| - \mathbf{d_{ij}})^2. \qquad (2)$$

In the context of manifold learning and dimensionality reduction, the MDS framework is enabled by estimating all pairwise geodesic distances with a shortest path algorithm like Dijkstra Dijkstra (1959) and using the minimizers of Equations 1 and 2 to generate global embeddings by preserving metric properties of the manifold.

Schwartz et al. (1989); Wolfson & Schwartz (1989) along with Tenenbaum et al. (2000) were the first to suggest populating the inter-geodesic matrix $\mathbf{D_s}$ using Dijkstra's algorithm for preserving metric properties of manifolds. The SMACOF algorithm of De Leeuw & Mair (2011) and their variants Bronstein et al. (2006) are iterative least squares scaling algorithms for minimizing Stress in Equation (2). The computational bottleneck of dealing with a dense distance matrix requiring all pairwise distances led to faster algorithms like Aflalo & Kimmel (2013); Shamai et al. (2015) and Boyarski et al. (2017), that used spectral representations of the Laplace-Beltrami Operator of the manifold. Rosman et al. (2010) shows a least squares scaling technique which can overcome holes and non-convex boundaries of the manifold. All these algorithms are non-parametric and few out-of-sample extensions have been suggested De Silva & Tenenbaum (2004) to generalize them to new samples.

## 2.3 Neural Networks for Manifold Learning

Examining the ability of neural networks to represent data manifolds has received considerable interest and has been studied from multiple perspectives. From the viewpoint of unsupervised parametric manifold learning, one notable approach is based on the metric-learning arrangement of the Siamese configuration Hadsell et al. (2006); Bromley et al. (1994); Chopra et al. (2005). Similarly, the parametric version of the Stochastic Neighborhood Embedding van der Maaten (2009) is another example of using a neural network to generate a parametric map that is trained to preserve local structure Maaten & Hinton (2008). However, these techniques demand an extensive training effort requiring large number of training examples in order to generate satisfactory embeddings. Weston et al. (2012) use a parametric network to learn a classifier which enforces a manifold criterion, requiring nearby points to have similar representations. Basri & Jacobs (2017) have argued that neural-networks can efficiently represent manifolds as monotonic chain of linear segments by providing an architectural construction and analysis. However, they do not address the manifold learning problem and their experiments are based on supervised settings where the ground-truth embedding is known a priori. Gong et al. (2006); Mishne et al. (2017); Chui & Mhaskar (2016) use neural networks specifically for solving the out-of-sample extension problem for manifold learning. However, their procedure involves training a network to follow a pre-computed non-parametric embedding rather than adopting an entirely unsupervised approach thereby inheriting some of the deficiencies of the non-parametric methods.

It is advantageous to adopt a parametric approach to non-linear dimensionality reduction and replace the computational block of the matrix construction and eigenvalue decomposition with a straightforward parametric computation. Directly characterizing the non-linear map provides a simple out-of-sample extension which is a plain forward pass of the network. More importantly, it is expected that the reduced and tightly controlled parameters would lead to an improved non-local generalization of the manifold Bengio et al. (2013).

In Hadsell et al. (2006) (henceforth DrLim), it was proposed to use Siamese Networks for manifold learning using the popular hinge-embedding criterion (Equation (3)) as a loss function. A Siamese configuration (Figure 2) comprises of two identical networks that process two separate units of data to achieve output pairs that are compared in a loss function. The contrastive training comprises of

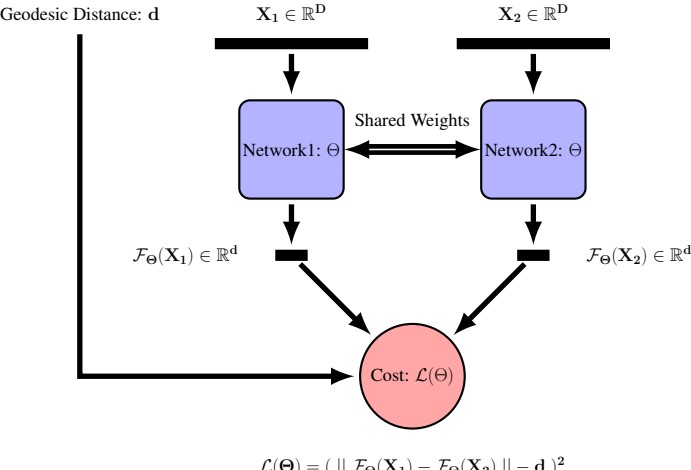

$$\mathcal{L}(\boldsymbol{\Theta}) = (\ ||\ \mathcal{F}_{\boldsymbol{\Theta}}(\mathbf{X_1}) - \mathcal{F}_{\boldsymbol{\Theta}}(\mathbf{X_2})\ ||\ -\ \mathbf{d}\ )^2$$

Figure 2: Siamese Configuration

constructing pairs $\{\mathbf{X_1^{(k)}}, \mathbf{X_2^{(k)}}, \lambda^{(\mathbf{k})}\}_{k=1}^{k=N}$ where $\lambda^{(k)} \in \{0, 1\}$ is a label for indicating a positive (a neighbor) or negative pair (not a neighbor) by building a nearest neighbor graph from the manifold data.

$$\mathcal{L}(\boldsymbol{\Theta}) = \sum_{\mathbf{k}} \lambda^{(\mathbf{k})}\ ||\mathcal{F}_{\boldsymbol{\Theta}}(\mathbf{X_1^{(k)}}) - \mathcal{F}_{\boldsymbol{\Theta}}(\mathbf{X_2^{(k)}})||$$
$$+(\mathbf{1} - \lambda^{(\mathbf{k})})\ \max\{\mathbf{0},\ \mu - ||\mathcal{F}_{\boldsymbol{\Theta}}(\mathbf{X_1^{(k)}}) - \mathcal{F}_{\boldsymbol{\Theta}}(\mathbf{X_2^{(k)}})||\} \quad (3)$$

Training with the loss in Equation (3) means that at any given update step, a negative pair contributes to the training only when their pairwise distance is less than $\mu$. This leads to a hard-negative sampling problem where the quality of embedding depends on the selection of negative examples in order to prevent excessive clustering of the neighbors. This typically requires an extensive training effort with a huge amount of training data (30000 positive and approximately 17.9 million negatives for a total of 6000 data samples as reported in Hadsell et al. (2006)).

## 3 A FRAMEWORK FOR PARAMETRIC MULTIDIMENSIONAL SCALING

We propose to incorporate the ideas of Least Squares Scaling into the computational infrastructure of the Siamese configuration as shown in Figure 2. For every $\mathbf{k}^{th}$ pair, we estimate the geodesic distance using a shortest path algorithm and train the network to preserve these distances by minimizing the Stress function

$$\mathcal{L}(\boldsymbol{\Theta}) = \sum_{\mathbf{k}} (\ ||\ \mathcal{F}_{\boldsymbol{\Theta}}(\mathbf{X_1^{(k)}}) - \mathcal{F}_{\boldsymbol{\Theta}}(\mathbf{X_2^{(k)}})\ ||\ - \mathbf{d^{(k)}}\ )^{\mathbf{2}} \quad (4)$$

The advantage of adopting the loss in Equation (4) over Equation (3) is that every pair of training data contributes to the learning process thereby eliminating the negative sampling problem. More importantly, it facilitates the use of efficient manifold sampling techniques like the Farthest Point Sampling strategy that make it possible to train with much fewer pairs of examples. The farthest point sampling strategy Bronstein et al. (2008) (also referred to as the MinMax strategy in De Silva & Tenenbaum (2004)) is a method to pick landmarks amongst the points of a discretely sampled manifold such that under certain conditions, these samples uniformly cover the manifold much as possible. Starting from a random selection, the landmarks are chosen one at a time such that, each new selection from the unused samples has the largest geodesic distance to the set of the selected sample points. Figure 3 provides a visualization of this sampling mechanism. We train the network by minimizing the loss in Equation (4) by computing the pairwise geodesic distances of the landmarks. Therefore, the pre-training computational effort is confined to computing the pairwise geodesic distances of only the landmark points. The proposed geometric manifold learning algorithm can be summarized in two steps,

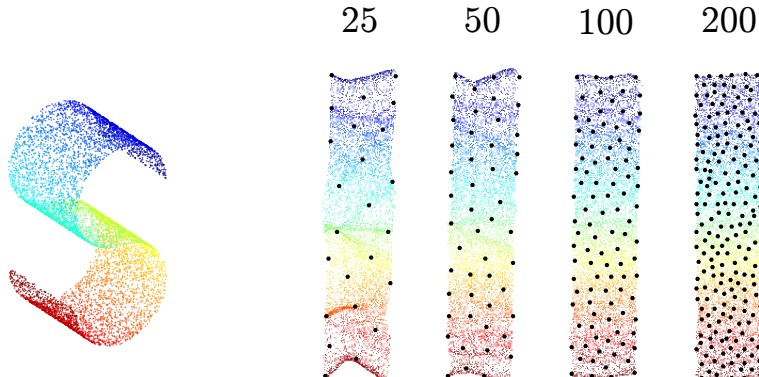

Figure 3: 2D embeddings of a three dimensional S-Curve Manifold generated by a 2-Layer MLP with 70 hidden nodes, that was trained with varying number of landmark points obtained with the Farthest Point Sampling Algorithm.

**Step1:** Compute the nearest-neighbor graph from the manifold data and obtain a set of landmark-points and their corresponding pairwise graph/geodesic distances using Dijkstra's algorithm and the Farthest Point Strategy.

**Step2:** Form a dataset of landmark pairs with corresponding geodesic distances $\{\mathbf{X}_1^{(\mathbf{k})}, \mathbf{X}_2^{(\mathbf{k})}, \mathbf{d}^{(\mathbf{k})}\}$ and train network in Siamese configuration using the least-squares MDS loss in Equation (4)

## 4 EXPERIMENTS

### 4.1 3D POINT CLOUD DATA

Our first set of experiments are based on point-cloud manifolds, e.g., like the Swiss Roll, S-Curve and the Helical Ribbon. We use a multilayer perceptron (MLP) with the PReLU() as the non-linear activation function, given by

$$PReLU(x) = \max(0, x) + a \ \min(0, x), \tag{5}$$

where $a$ is a learnable parameter. The networks are trained using the ADAM optimizer with constants $(\beta_1, \beta_2) = (0.95, 0.99)$ and a learning rate of 0.01 for 1000 iterations. We run each optimization 5 times with random initialization to ensure convergence. All experiments are implemented in python using the PyTorch framework Paszke et al. (2017). We used the scikit-learn machine learning library for the nearest-neighbor and scipy-sparse for Dijkstra's shortest path algorithms.

Figure 1 and 3 show the results of our method on the Helical Ribbon and S-Curve respectively with varying number of training samples (in black) out of a total of 8172 data points. The number of landmarks dictates the approximation quality of the low-dimensional embedding generated by the network. Training with too few samples will result in inadequate generalization which can be inferred from the corrugations of the unfurled results in the first two parts of Figure 3 and increasing the number of landmarks expectedly improves the quality of the embedding. We compute the Stress function (Equation (2)) of the entire point configuration to measure the quality of the MDS fit. Figure 4a shows the decay in the Stress as a function of the number of training points (or Landmarks) of a 2-Layer MLP.

The natural next question to ask is *how many landmarks? how many layers? and how many hidden nodes per layer?* We observe that these questions relate to an analogous setup in numerical methods for differential equations. For a given numerical technique, the accuracy of the solution depends on the resolution of the spatial grid over which the solution is estimated. Therefore, numerical methods are ranked by an assessment of the *order of accuracy* their solutions observe. This can be obtained

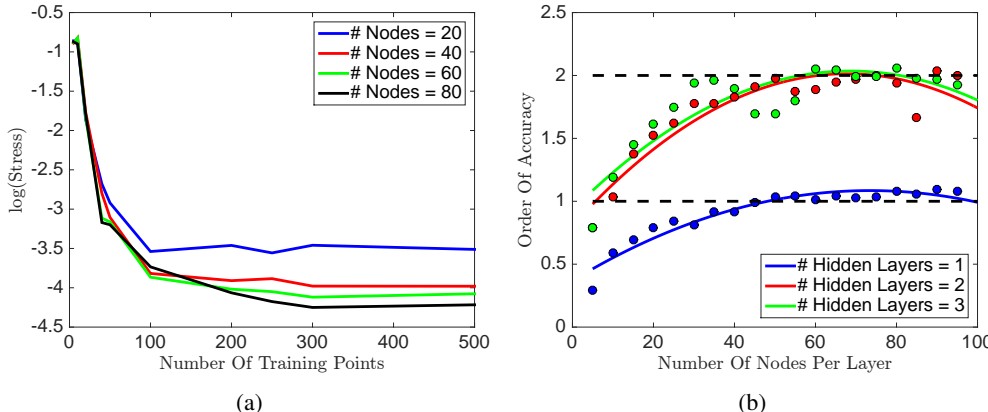

(a)                                                     (b)

Figure 4: **Exploring the variations in architecture and the number of landmarks:** (a) The logarithm of stress of all 8172 points as a function of number of landmarks. (b) Order of accuracy estimates of varying architectures.

by assuming that the relationship between the approximation error $E$ and the resolution of the grid $h$ is given by

$$E = C\ h^P \tag{6}$$

where $P$ is the order of accuracy of the technique and $C$ is some constant. Therefore, $P$ is obtained by computing the slope of the line obtained by charting $\log(E)$ vs $\log(h)$,

$$\log(E) = \log(C) + P\ \log(h). \tag{7}$$

We extend the same principle to evaluate network architectures (in place of numerical algorithms) for estimating the quality of isometric maps. We use the generalized Stress in Equation (2) as the error function for Equation (6). We assume that due to the 2-approximate property of the farthest point strategy Hochbaum & Shmoys (1985) the sampling is approximately uniform and hence $h \propto \frac{1}{\sqrt{K}}$ where $K$ is the number of landmarks. By varying the number of layers and the number of nodes per layer we associate an order of accuracy to each architecture using Equation (7). Figure 4b shows the results of our experiment. It shows that a single layer MLP has the capacity for modeling functions upto the first order of accuracy. Adding an additional layer increases the representational power by moving to a second order result. Adding more layers does not provide any substantive gain arguably due to a larger likelihood of over-fitting as seen in the considerably noisier estimates (in green). Therefore, a two layer MLP with 70 hidden nodes per layer can be construed as a good architecture for approximating the isometric map of the S-Curve of Figure 3 with 200 landmarks.

### 4.2 IMAGE ARTICULATION MANIFOLDS

We extend the parametric MDS framework to image articulation manifolds where each sample point is a binary image governed by the modulation of a few parameters. We specifically deal with image manifolds that are isometric to Euclidean space Donoho & Grimes (2005), that is, the geodesic distance between any two sample points is equal to the euclidean distance between their articulation parameters. In the context of the main discussion of this paper, which is metric preserving properties of manifolds, we find that such datasets provide an appropriate test-bed for evaluating metric preserving algorithms.

We construct a horizon articulation manifold where each image contains two distinct regions separated by a horizon which is modulated by a linear combination of two fixed sinusoidal basis elements. See Figure 5.

$$
\begin{aligned}
I_{\alpha_1,\alpha_2}(u,v) &= \mathbb{1}_{\{v \le \psi_{\alpha_1,\alpha_2}(u)\}} \\
\psi_{\alpha_1,\alpha_2}(u) &= \alpha_1 \sin(\omega_1 u) + \alpha_2 \sin(\omega_2 u)
\end{aligned}
\tag{8}
$$

Thus, each sample has an intrinsic dimensionality of two - the articulation parameters $(\alpha_1, \alpha_2)$ which govern how the sinusoids representing the horizon are mixed. We sample the articulation

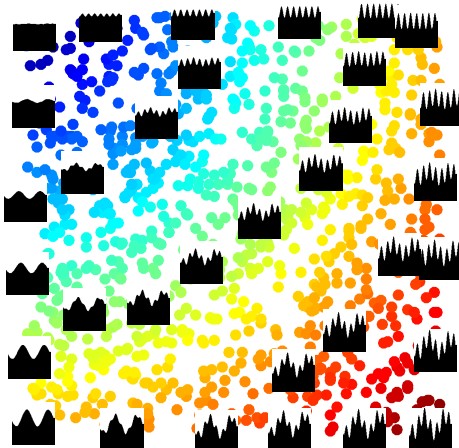

Figure 5: **Visualizing a horizon articulation manifold:** 1000 samples generated as per equations 8 and 9 with $\omega_1 = 2, \omega_2 = 7$. The color is proportional to the magnitude $\sqrt{\alpha_1^2 + \alpha_2^2}$

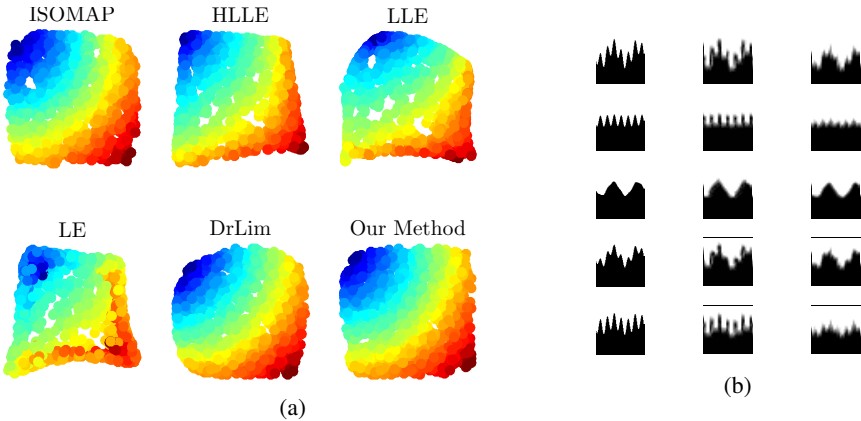

Figure 6: (a) Comparing metric preserving properties for different manifold learning algorithms on the image articulation manifold dataset. LE: Belkin & Niyogi (2003), DrLim: Hadsell et al. (2006). The proposed method shows maximum fidelity to the ground truth shown in Figure 5 (b) Visualizing the outputs of some filters trained using our method. The 1st column shows the input images. The filters act to exaggerate (2nd column) and suppress (3rd column) a governing frequency in Equation (8).

parameters from a 2D uniform distribution

$$(\alpha_1, \alpha_2) \sim U([0, 1] \times [0, 1]). \tag{9}$$

We generate 1000 images of the horizon articulation manifold of size $100 \times 100$. The network architecture comprises of two convolution layers each with kernel sizes 12 and 9, number of kernels 15 and 2 respectively along with a stride of 3 and followed by a fully connected layer mapping the image to a two dimensional entity. We train using the ADAM optimizer Kingma & Ba (2015) with a learning rate of 0.01 and parameters $(\beta_1, \beta_2) = (0.95, 0.99)$. We train the network using 50 Landmark points.

Figure 6a shows the comparison between our method and other non-parametric counterparts along with the parametric approach of DrLim. Training with the least squares loss of Equation (4) shows high fidelity to the ground truth of Figure 5. Except for Isomap and our proposed method, all other methods show some form of distortion indicating a suboptimal metric preservation property.

Training with the articulation manifold in Figure5 provides an opportunity to get more detail in understanding the parametric action of the neural networks. The 2nd and 3rd columns in Figure 6b show the outputs of some of the filters in the first layer of the architecture trained on the manifold in Figure 5. The distance preserving loss facilitates the learning of image filters which separate the underlying frequencies governing the non-linear manifold, thereby providing a visual validation of the parametric map.

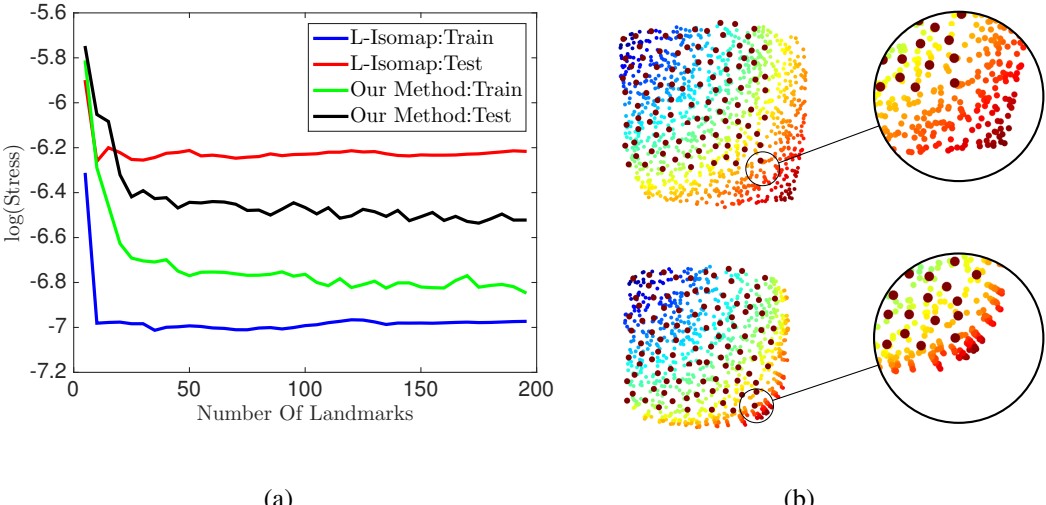

(a)                         (b)

Figure 7: **Comparing with nonparametric out-of-sample extensions:** (a) Comparing numerical stress values of embeddings generated for training and test data of our method and the interpolating formulation of Landmark Isomap. (b) Visualizing the non-local generalization properties of our method (top) and Landmark Isomap (bottom). Both algorithms were trained on the same Landmarks (in red) sampled from only a part of the manifold.

## 4.3 COMPARISON WITH LANDMARK ISOMAP

We compare our parametric multi-dimensional scaling approach to its direct non-parametric competitor: Landmark-Isomap De Silva & Tenenbaum (2004). The main idea of Landmark-Isomap is to perform classical scaling on the inter-geodesic distance matrix of only the landmarks and to estimate the embeddings of the remaining points using an interpolating formula (also mentioned in Bengio et al. (2004)). The formula uses the estimated geodesic distance of each new point to the selected Landmarks in order to estimate its low dimensional embedding.

We use the image articulation manifold dataset to provide a quantitative and visual comparison between the two methods. Both the methods are imputed with the same set of landmarks for evaluation. In the first experiment, we generate two independent horizon articulation datasets each containing 1000 samples generated using Equations (8) and (9) for training and testing. We then successively train both algorithms on the training dataset with varying number of Landmark points and then use the examples from the test data-set to evaluate performance. Figure 7 (a) shows that low dimensional embedding of Landmark-Isomap admits smaller stress values (hence better metric preservation) for the training dataset, but behaves poorly on unseen examples. On the other hand, despite larger stress values for training, the network shows better generalizability.

In order to visualize *non-local* generalization properties, we repeat the previous experiment with a minor modification. We train both algorithms on horizon articulation manifolds with parameters sampled from $(\alpha_1, \alpha_2) \sim U([0, 0.75] \times [0, 0.75])$ and visualize the outputs on test datasets with parameters sampled from $(\alpha_1, \alpha_2) \sim U([0, 1] \times [0, 1])$ thereby isolating a part of the manifold during training. As shown in Figure 7b, the output of Landmark-Isomap shows a clustered result due to the lack of non-local data in the geodesic distance calculations for the interpolation. In contrast, the network clearly shows a better generalization property.

## 4.4 CAMERA POSE MANIFOLDS

Finally we test our method on a more realistic dataset where the constraint of being isometric to a low dimensional euclidean space is not necessarily strict. We generate 1369 images obtained by smoothly varying the azimuth and elevation of the camera.

Figure 8 shows the embeddings and associated training times of our method compared to the *conceptual* (Landmark Isomap) and *computational* (DrLim Hadsell et al. (2006)) siblings. We see that

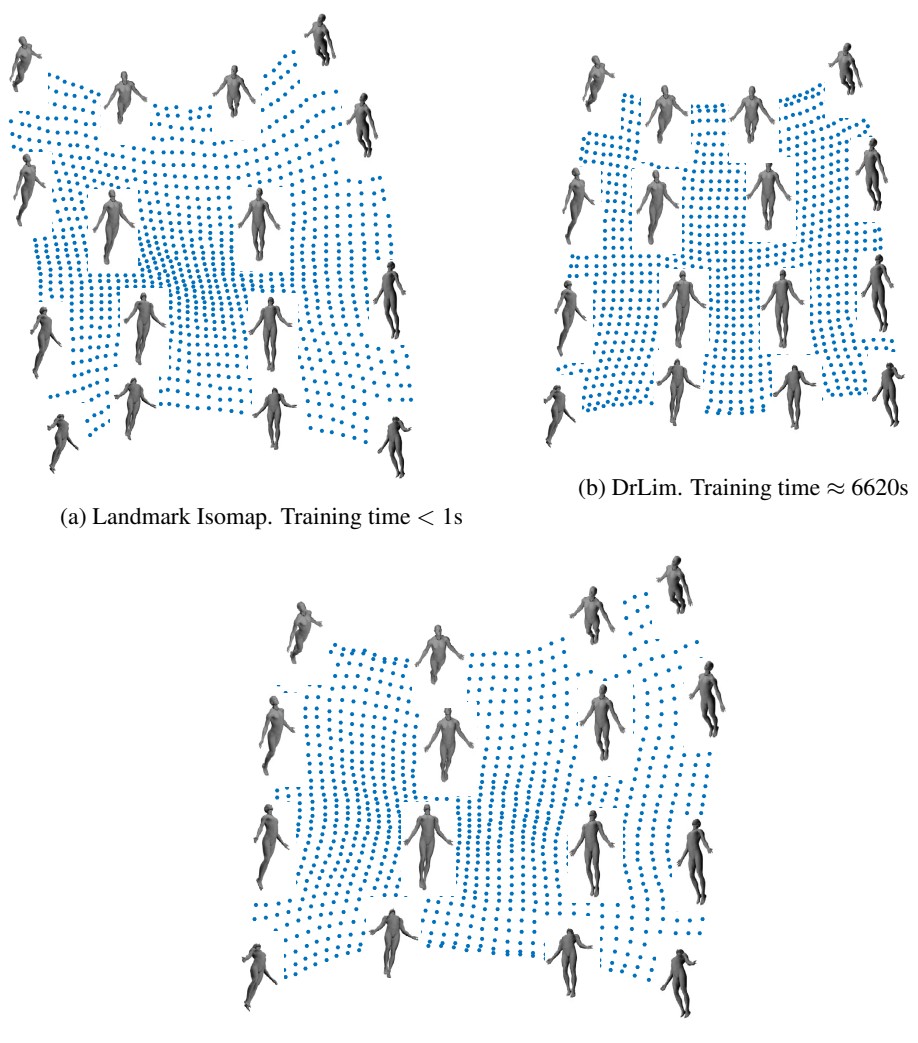

(a) Landmark Isomap. Training time $< 1s$

(b) DrLim. Training time $\approx 6620s$

(c) Our Method. Training time $\approx 860s$

Figure 8: **Camera Pose Manifold:** Embedding results and training times for Landmark-Isomap, DrLim Hadsell et al. (2006) and our approach. Our hybrid parametric approach shows a faithful result for a smaller training time.

the fast interpolation scheme of Landmark Isomap demonstrates higher distortion whereas DrLim requires a considerable training effort, requiring all possible $\binom{1369}{2} = 936396$ pairs. Our integrated approach yields an improved result in a considerably smaller training time (geodesic distances between only $\binom{600}{2} = 179700$ pairs). We used the same 600 landmarks from Isomap and the same architecture of DrLim for generating the embedding in Figure 8c.

## 5 CONCLUSION

In the interest of obtaining a better understanding of neural network behavior, we advocate using learning methodologies for solving geometric problems with data by allowing a limited infusion of axiomatic computation to the learning process. In this paper we demonstrate such a scheme by combining parametric modeling with neural networks and the geometric framework of multidimensional scaling. The result of this union leads to reduction in training effort and improved local and non-local generalization abilities. As future work, we intend to further explore methods that leverage learning methodologies for improving the largely axiomatic setups of numerical algorithms.

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
