# OpenReview forum: "Parametric Manifold Learning Via Sparse Multidimensional Scaling"
_ICLR.cc/2018/Conference — Reject_

### Official Review · AnonReviewer3 · 2017-11-23
**The paper proposes a parametric manifold learning method based on deep learning and Siamese networks. Key references are missing (SAMMANN, auto-encoders); Experiments are limited.**

**Rating:** 5
**Confidence:** 5

**Review:**

The paper describes a manifold learning method that adapts the old ideas of multidimensional scaling, with geodesic distances in particular, to neural networks. The goal is to switch from a non-parametric to a parametric method and hence to have a straightforward out-of-sample extension.

The paper has several major shortcomings:
* Any paper dealing with MDS and geodesic distances should test the proposed method on the Swiss roll, which has been the most emblematic benchmark since the Isomap paper in 2000. Not showing the Swiss roll would possibly let the reader think that the method does not perform well on that example. In particular, DR is one of the last fields where deep learning cannot outperform older methods like t-SNE. Please add the Swiss roll example.
* Distance preservation appears more and more like a dated DR paradigm. Simple example from 3D to 2D are easily handled but beyond the curse of dimensionality makes things more complicated, in particular due to norm computation. Computation accuracy of the geodesic distances in high-dimensional spaces can be poor. This could be discussed and some experiments on very HD data should be reported.
* Some key historical references are overlooked, like the SAMMANN. There is also an over-emphasis on spectral methods, with the necessity to compute large matrices and to factorize them, probably owing to the popularity of spectral DR metods a decade ago. Other methods might be computationally less expensive, like those relying on space-partitioning trees and fast multipole methods (subquadratic complexity). Finally, auto-encoders could be mentioned as well; they have the advantage of providing the parametric inverse of the mapping too.
* As a tool for unsupervised learning or exploratory data visualization, DR can hardly benefit from a parametric approach. The motivation in the end of page 3 seems to be computational only.
* Section 3 should be further detailed (step 2 in particular).
* The experiments are rather limited, with only a few artifcial data sets and hardly any quantitative assessment except for some monitoring of the stress. The running times are not in favor of the proposed method. The data sets sizes are, however, quite limited, with N<10000 for point cloud data and N<2000 for the image manifold.
* The conclusion sounds a bit vague and pompous ('by allowing a limited infusion of axiomatic computation...'). What is the take-home message of the paper?

---

> ### Author Response · Authors · 2017-12-31
> **Reply To Reviewer 3**
>
> -Any paper dealing with MDS and geodesic distances should test the proposed method on the Swiss roll, which has been the most emblematic benchmark since the Isomap paper in 2000. Not showing the Swiss roll would possibly let the reader think that the method does not perform well on that example. In particular, DR is one of the last fields where deep learning cannot outperform older methods like t-SNE. Please add the Swiss roll example.
>
> Our framework works for swiss roll dataset as well. However, since the S-Curve is a similar dataset, we do not see merit to include it.
>
> -Distance preservation appears more and more like a dated DR paradigm. Simple example from 3D to 2D are easily handled but beyond the curse of dimensionality makes things more complicated, in particular due to norm computation. Computation accuracy of the geodesic distances in high-dimensional spaces can be poor. This could be discussed and some experiments on very HD data should be reported.
>
> We agree with this assessment. However, the main message of our paper was the parameterization of maps that preserve these geodesic distances. We highlight that this leads to improved local and non-local generalization abilities and allows us to analyze neural networks using tools from numerical analysis. We agree that estimation of geodesic distances is indeed a problem when dimensionality is very high, however in this paper our message is: if they are available, our parametric framework provides for an interesting performance analysis along with showing benefit in terms of computation and performance.
>
> -Some key historical references are overlooked, like the SAMMANN. There is also an over-emphasis on spectral methods, with the necessity to compute large matrices and to factorize them, probably owing to the popularity of spectral DR metods a decade ago. Other methods might be computationally less expensive, like those relying on space-partitioning trees and fast multipole methods (subquadratic complexity). Finally, auto-encoders could be mentioned as well; they have the advantage of providing the parametric inverse of the mapping too.
>
> Thank you for this input. We will add the references.
>
> -As a tool for unsupervised learning or exploratory data visualization, DR can hardly benefit from a parametric approach. The motivation in the end of page 3 seems to be computational only.
>
> We do not completely agree. Apart from showing a scheme with considerably reduced training effort, As Figure 7 demonstrates, parametric approaches show improved local and non-local generalization abilities.
>
> -* Section 3 should be further detailed (step 2 in particular).
>
> Thank you for the input. We will elaborate.
>
> -The experiments are rather limited, with only a few artifcial data sets and hardly any quantitative assessment except for some monitoring of the stress. The running times are not in favor of the proposed method. The data sets sizes are, however, quite limited, with N<10000 for point cloud data and N<2000 for the image manifold.
>
> We are currently exploring more richer and larger datasets. However, in the interest of studying network behavior under strict metric constraints we found it appropriate to test on the articulation dataset whose geometry is well understood.
>
> -The conclusion sounds a bit vague and pompous ('by allowing a limited infusion of axiomatic computation...'). What is the take-home message of the paper?
>
> We will rephrase this statement. What we wish to highlight in this paper is that it is useful to reformulate classical algorithms like multidimensional scaling using a parametric approach with neural networks. Why? Because  techniques like geodesic sampling methods can be leveraged in order to (1.) Evaluate network architectures and characterize them using numerical constructs like order of accuracy and (2.) Obtain improved local and non-local generalization abilities in comparison to previous kernel extrapolation techniques minimizing the same objective.

---

### Official Review · AnonReviewer1 · 2017-11-27
**many small issues and not much novelty**

**Rating:** 4
**Confidence:** 4

**Review:**

The authors argue that the spectral dimensionality reduction techniques are too slow, due to the complexity of computing the eigenvalue decomposition, and that they are not suitable for out-of-sample extension. They also note the limitation of neural networks, which require huge amounts of data to properly learn the data structure. The authors therefore propose to first sub-sample the data and afterwards to learn an MDS-like cost function directly with a neural network, resulting in a parametric framework.

The paper should be checked for grammatical errors, such as e.g. consistent use of (no) hyphen in low-dimensional (or low dimensional).

The abbreviations should be written out on the first use, e.g. MLP, MDS, LLE, etc.

In the introduction the authors claim that the complexity of parametric techniques does not depend on the number of data points, or that moving to parametric techniques would reduce memory and computational complexities. This is in general not true. Even if the number of parameters is small, learning them might require complex computations on the whole data set. On the other hand, even if the number of parameters is equal to the number of data points, the computations could be trivial, thus resulting in a complexity of O(N).

In section 2.1, the authors claim "Spectral techniques are non-parametric in nature"; this is wrong again. E.g. PCA can be formulated as MDS (thus spectral), but can be seen as a parametric mapping which can be used to project new words.

In section 2.2, it says "observation that the double centering...". Can you provide a citation for this?

In section 3, the authors propose they technique, which should be faster and require less data than the previous methods, but to support their claim, they do not perform an analysis of computational complexity. It is not quite clear from the text what the resulting complexity would be. With N as number of data points and M as number of landmarks, from the description on page 4 it seems the complexity would be O(N + M^2), but the steps 1 and 2 on page 5 suggest it would be O(N^2 + M^2). Unfortunately, it is also not clear what the complexity of previous techniques, e.g DrLim, is.

Figure 3, contrary to text, does not provide a visualisation to the sampling mechanism.

In the experiments section, can you provide a citation for ADAM and explain how the parameters were selected? Also, it is not meaningful to measure the quality of a visualisation via the MDS fit. There are more useful approaches to this task, such as the quality framework [*].

In figure 4a, x-axis should be "number of landmarks".

It is not clear why the equation 6 holds. Citation?
It is also not clear how exactly the equation 7 is evaluated. It says "By varying the number of layers and the number of nodes...", but the nodes and layer are not a part of the equation.

The notation for equation 8 is not explained.

Figure 6a shows visualisations by different techniques and is evaluated "by looking at it". Again, use [*].

[*] Lee, John Aldo ; Verleysen, Michel. Scale-independent quality criteria for dimensionality reduction. In: Pattern Recognition Letters, Vol. 31, no. 14, p. 2248-2257 (2010). doi:10.1016/j.patrec.2010.04.013.

---

> ### Author Response · Authors · 2017-12-31
> **Reply to Reviewer 1**
>
> We thank the reviewer on his feedback.
>
> Issues on complexity:
>
> - In the introduction the authors claim that the complexity of parametric techniques does not depend on the number of data points, or that moving to parametric techniques would reduce memory and computational complexities. This is in general not true. Even if the number of parameters is small, learning them might require complex computations on the whole data set. On the other hand, even if the number of parameters is equal to the number of data points, the computations could be trivial, thus resulting in a complexity of O(N).
>
> - In section 3, the authors propose they technique, which should be faster and require less data than the previous methods, but to support their claim, they do not perform an analysis of computational complexity. It is not quite clear from the text what the resulting complexity would be. With N as number of data points and M as number of landmarks, from the description on page 4 it seems the complexity would be O(N + M^2), but the steps 1 and 2 on page 5 suggest it would be O(N^2 + M^2). Unfortunately, it is also not clear what the complexity of previous techniques, e.g DrLim, is.
>
> What we wanted to highlight was that training a network to preserve geodesic distances shows computational benefits as compared to performing large-scale eigendecompositions having the same principle. However such an analysis is not available for the other network based methods: DrLim and Parametric t-SNE and  possibly merits a separate paper, dedicated to this line of investigation.
>
> -It is not meaningful to measure the quality of a visualisation via the MDS fit. There are more useful approaches to this task, such as the quality framework [*].
>
> Thank you for this input. However, since we show comparison only for a specific set of algorithms (MDS) we found it very straightforward most appropriate to compare with the Stress function which measures how well manifold distances have been preserved for all points, especially when we train only on a subset of points.
>
> -It is not clear why the equation 6 holds. Citation?
> It is also not clear how exactly the equation 7 is evaluated. It says "By varying the number of layers and the number of nodes...", but the nodes and layer are not a part of the equation.
>
> This is a standard analysis performed for finite difference schemes used for solving differential equations. For citation see: LeVeque, Randall J. "Finite difference methods for differential equations." Draft version for use in AMath 585.6 (1998). Basically numerical schemes estimate some function (for e.g. the solution to a differential equation) over a given number of sampled points on its domain. Therefore, the order of accuracy of the scheme is estimated by using Equation 7 after plotting the approximation error of the scheme as a function of the grid spacing (or indirectly the number of sampled points) and extracting the slope of this line.
>
> We employ this same philosophy in our problem with a slight modification. We employ the global stress of Equation 2 as our error function (since a non-zero stress corresponds to a suboptimal solution as far as distance preserving algorithms are concerned).  A specific network (with a given number of hidden layers and nodes per layer) corresponds to a numerical scheme and hence determines a unique error profile (like Fig 4a) and hence can be graded with the order of accuracy it demonstrates in unfurling a manifold like the S-Curve. By varying the components of the network (number of layers and hidden nodes per layer) , we obtain different error profiles (the term ‘E’ in equations 6 & 7) and hence different order of accuracies corresponding to each choice of the parameters.
>
> -Figure 6a shows visualisations by different techniques and is evaluated "by looking at it". Again, use [*].
>
> Since we know the geometry of the manifold (the articulation data is perfectly isometric as proved in: Donoho, David L., and Carrie Grimes. "Image manifolds which are isometric to Euclidean space." Journal of mathematical imaging and vision 23.1 (2005): 5-24.), a visual validation of Figure 6 clearly shows which algorithms have the best metric preservation.

---

### Official Review · AnonReviewer2 · 2017-11-27
**Incremental improvement of an old manifold learning algorithm**

**Rating:** 3
**Confidence:** 4

**Review:**

The key contribution of the paper is a new method for nonlinear dimensionality reduction.

The proposed method is (more or less) a modification of the DrLIM manifold learning algorithm (Hadsell, Chopra, LeCun 2006) with a slightly different loss function that is inspired by multidimensional scaling. While DrLIM only preserves local geometry, the modified loss function presents the opportunity to preserve both local and global geometry. The rest of the paper is devoted to an empirical validation of the proposed method on small-scale synthetic data (the familiar Swiss roll, as well as a couple of synthetic image datasets).

The paper revisits mostly familiar ideas. The importance of preserving both local and global information in manifold learning is well known, so unclear what the main conceptual novelty is. This reviewer does not believe that modifying the loss function of a well established previous method that is over 10 years old (DrLIM) constitutes a significant enough contribution.

Moreover, in this reviewer's experience, the major challenge is to obtain proper estimates of the geodesic distances between far-away points on the manifold, and such an estimation is simply too difficult for any reasonable dataset encountered in practice. However, the authors do not address this, and instead simply use the Isomap approach for approximating geodesics by graph distances, which opens up a completely different set of challenges (how to construct the graph, how to deal with "holes" in the manifold, how to avoid short circuiting in the all-pairs shortest path computations etc etc).

Finally, the experimental results are somewhat uninspiring. It seems that the proposed method does roughly as well as Landmark Isomap (with slightly better generalization properties) but is slower by a factor of 1000x.

The horizon articulation data, as well as the pose articulation data, are both far too synthetic to draw any practical conclusions.

---

> ### Author Response · Authors · 2017-12-31
> **Reply to Reviewer 2**
>
> We thank the reviewer on his feedback.
>
> The main message of our paper is to show a merge between classical geometric algorithms and parametric learning methodologies.  Although it is true that both Multidimensional Scaling and DrLim are indeed well established, we felt it is important to connect the two and show the advantages in adopting a parametric approach to a classical algorithm like multidimensional scaling. To this aid we demonstrate two experiments: (1.) the use of geodesic sampling methods which allow one to obtain a numerical assessment of the network architecture and show a substantive reduction in training effort as compared to DrLim and (2.) Providing a qualitative and quantitative comparison with the analogous nonparametric out-of-sample techniques.
>
> We believe that there is no case to be made for a “best” manifold learning algorithm that works on all types of data. Every dataset will have a corresponding algorithm whos primary design principles will best suit it and therefore we do not claim we have a method which works universally. What we have said in this paper is this: In the class of methods that preserve geodesic distances, we find substantive merit to adopt a parametric approach by using a neural network. Why? because the network unambiguously shows better local and non-local generalization abilities on manifold datasets whose geometry is very clearly understood and also the fact that we can gauge how good the network performs by estimating it's order of accuracy. We believe that the novelty of our work lies in the analysis rather than the solution itself (which as rightly argued by the reviewer is well known). Therefore our choice of using a synthetic dataset like the image articulation manifold was to aid a more clear analysis of manifold learning performance. For the class of algorithms we have claimed to better, this performance is best encapsulated by the Stress function.

---

### Decision · Program_Chairs · 2018-01-29
**ICLR 2018 Conference Acceptance Decision**

**Decision:**

Reject

**Comment:**

Dear authors,

Thank you for your submission to ICLR. Sadly, the reviewers were not convinced by the novelty of your approach nor by its experimental results. Thus, your paper cannot be accepted to ICLR.